# Soluble Transferrin Receptor, Antioxidant Status and Cardiometabolic Risk in Apparently Healthy Individuals

**DOI:** 10.3390/antiox12010019

**Published:** 2022-12-22

**Authors:** Milton Fabian Suárez-Ortegón, Alejandra Arbeláez, José María Moreno-Navarrete, José Guillermo Ortega-Ávila, Mildrey Mosquera, José Manuel Fernández-Real

**Affiliations:** 1Departamento de Alimentación y Nutrición, Facultad de Ciencias de La Salud, Pontificia Universidad Javeriana Seccional Cali, Cali 760030, Colombia; 2Nutrition Group, Universidad del Valle, Cali 760030, Colombia; 3Physiological Sciences Department, Universidad del Valle, Cali 760030, Colombia; 4Department of Diabetes, Endocrinology and Nutrition, Institut d’Investigació Biomèdica de Girona (IdIBGi), 28029 Madrid, Spain; 5Departamento de Ciencias Básicas, Facultad de Ciencias de La Salud, Pontificia Universidad Javeriana Seccional Cali, Cali 760030, Colombia; 6CIBEROBN (CB06/03/010), Instituto de Salud Carlos III (ISCIII), 28029 Madrid, Spain

**Keywords:** transferrin receptor, cardiometabolic risk, antioxidant status, iron

## Abstract

Body iron excess appears to be related to insulin resistance and cardiometabolic risk and increased oxidative stress might be involved in this relationship. Very few studies have described the association between soluble transferrin receptor (sTfR) levels and cardiometabolic risk in the general population or antioxidant status. There were 239 subjects (20–65 years old) included in this cross-sectional study. Linear regressions adjusting for BMI, menopausal status, insulin resistance (HOMA-IR), physical inactivity, alcohol intake and subclinical/chronic inflammation were used to describe the association between sTfR, total antioxidant capacity (TAC), and measures of cardio-metabolic risk. sTfR levels were positively associated with TAC in men (βeta [95% confidence interval ]: 0.31 [0.14 to 0.48]) and women (βeta = 0.24 [0.07 to 0.40]) in non-adjusted and adjusted models (*p* < 0.05). In men, sTfR levels were inversely associated with waist circumference (βeta [95% confidence interval]: −1.12 [−2.30 to −0.22]) and fasting glucose (−2.7 (−4.82 to −0.57), and positively with LDL cholesterol (12.41 (6.08 to 18.57) before and after adjustments for confounding variables. LDL cholesterol had a significant and positive association with TAC in non-adjusted and adjusted models in men (*p* < 0.05). sTfR levels are significantly associated with antioxidant status and a few specific cardio-metabolic risk variables, independently of covariates that included serum ferritin and hepcidin. This might imply that iron biomarkers in regard to cardiometabolic risk reflect physiological contexts other than iron metabolism.

## 1. Introduction

Following the description of cardiovascular disease and type 2 diabetes as common complications in iron overload states such as transfusion-treated thalassemia [1,2], several studies have explored the relationship between iron levels and cardio-metabolic risk in the general populations. Body iron stores, estimated using serum ferritin levels, have been the most widely used parameter. Associations of serum ferritin with insulin resistance and cardio-metabolic diseases have been widely reported [3,4,5]. Increased oxidative stress derived from pro-oxidant properties of iron results in impaired insulin signaling as a possible mechanism for these relationships [6].

While some studies have identified associations between serum ferritin and the metabolic syndrome [4], less is known about the relationship between other markers of iron metabolism such as the soluble transferrin receptor (sTfR) and cardio-metabolic risk and the possible role of oxidative stress in this relationship. sTfR levels are proportional to the cellular expression of the membrane-associated transferrin receptors that accurately reflect cellular iron demands, and therefore, the higher the body iron stores, the lower the sTfR levels [7].

The findings about relationship between type 2 diabetes and sTfR in previous studies are conflicting according to a recent meta-analysis [8]. Some studies that have considered basic adjustments for confounding factors [9,10] have reported negative results. Since sTfR could be modulated by pleiotropic effects [11], it is still not clear whether the possible associations with cardio-metabolic risk might be similar to those observed for ferritin and whether these possible associations are independent of adiposity, insulin resistance or markers of subclinical inflammatory activity, and even independent of other iron metabolism markers. In the limited studies conducted to date, robust multivariate analyses have not been approached to clarify whether independent associations exist. In addition, there is very little information on this relationship in Latin-American populations in which it is important to describe associations patterns.

Taking into account the aspects mentioned above, we aimed to investigate the associations of circulating sTfR levels with antioxidant status and variables related to cardiometabolic risk before and after adjusting for potential confounding variables in apparently healthy individuals from Southwest Colombia.

## 2. Materials and Methods

The study population consisted of 239 volunteers from the staff of a hospital, a university, a governmental health department and a supermarket chain in Cali-Colombia, who responded to advertisements describing our study. In order to obtain healthy subjects and avoid bias in estimating variables related to cardiometabolic risk, total antioxidant capacity (TAC) and iron parameters, the following exclusion criteria were considered: Clinically significant diseases in the liver, neurologic or endocrine systems, cardiometabolic diseases (hypertension, history of stroke, myocardial infarction, or type 2 diabetes) or other major systemic disease; smoking; blood transfusion or iron therapies during the previous six months; long-term multivitamin or vitamin supplements consumption (two or more days/week); medication to lower lipid or glycaemia levels; current evidence of acute or chronic inflammatory or infective diseases; and history of disturbances in iron balance (e.g., hemosiderosis from any cause, hemolytic anemia, iron deficiency). The study sample has a serum bank preserved at −70 Celsius grades in which additional biochemical measurements have been progressively conducted via new funding and collaborations. Previous research findings on cardiometabolic risk in relation to other iron markers such as serum ferritin and hepcidin have been published [3,12] The Universidad del Valle Research Ethics Committee approved the study (Permission number: 0016-07), and all participants gave written informed consent.

### 2.1. Clinical Measurements

Blood pressure was measured using digital sphygmomanometers with an appropriately sized cuff in a sitting position after 15 min rest. The measurement was repeated after five minutes. The mean of the two measurements was used in the statistical analyses. Body weight and height were measured using standard techniques and instruments and body mass index (BMI) was calculated as weight in kg/height in meters^2^. Waist circumference (WC) was measured from the midpoint between the lateral iliac crest and the lowest rib using a flexible steel tape measure. A survey to record personal data, life-style habits were recorded by trained interviewers.

### 2.2. Biochemical Measurements

Body weight and height were measured using standard techniques and instruments and body mass index (BMI) was then calculated. Fasting glucose, triglycerides, total cholesterol and lipoprotein cholesterol (HDL-C) were determined by using enzymatic-colorimetric assays (Biosystems Inc., Spain); Low-density cholesterol (LDL-C) levels were calculated according to Friedewald equation: total cholesterol-(HDL-C + (triglycerides/5) [13]. Serum ferritin and high sensitivity C reactive protein (CRP) were measured using turbidimetry (Biosystems, Spain). Fasting insulin was measured using quimioluminiscence. Levels of hepcidin and soluble transferrin receptor (sTfR) were measured using double monoclonal sandwich enzyme immunoassays [DRG^®^ Hepcidin 25 (Bioactive) ELISA (EIA-5258, DRG International, Inc., New Jersey, NY, USA); and Human sTfR ELISA (RD194011100, Heidelberg, Germany), respectively]. Intra- and interassay coefficients of variation were <5.5%. HOMA-IR (Homeostatic Model Assessment Insulin resistance) was calculated as insulin mU /mL x glucose mg-dL/405) [14]. Total antioxidant capacity (TAC) was measured using a colorimetric assay (Cayman chemical, Cat No.709001, Ann Arbor MI) which assesses the total antioxidant capacity of serum in terms of the ability of antioxidants in the sample to inhibit the oxidation of ABTS^®^ (2,2’-azino-di-[3-ethylbenzthiazoline sulphonate]) to ABTS^®^+by metmyoglobin. The capacity of the serum antioxidants is compared to Trolox, a tocopherol analogue, and quantified as molar Trolox equivalents.

### 2.3. Statistical Analysis

All the analyses were conducted in each gender and the study variables were described as means and their standard deviation or median and their interquartile range according to the distribution of variables. Differences were estimated via t-student test or Mann–Whitney U test. Multivariate linear regression analysis was conducted to evaluate and adjust the associations of sTfR with cardiometabolic risk factors and TAC. Regression coefficients were described as non-adjusted and adjusted for age, CRP levels, BMI, alcohol intake (no/yes), physical inactivity (as none physical activity per week), HOMA-IR and other iron metabolism markers (levels of ferritin and hepcidin). Analyses were performed on transformed values for skewed variables: Logarithm of sTfR, systolic blood pressure, triglycerides, HOMA-IR, and ferritin, and 1/hs-CRP. A *p* value <0.05 was considered statistically significant. Analyses were performed using STATA 8.0.

## 3. Results

Clinical and biochemical characteristics of participants are described in Table 1. Men and women were similar in age, BMI and total antioxidant capacity (TAC). As expected, men had lower levels of sTfR and HDL-C, and higher values of waist circumference, blood pressure, and serum ferritin and hepcidin than women. Women showed higher levels of CRP than men.

Serum sTfR concentration was positively associated with TAC in both genders, and this association remained significant after adjustments (Figure 1). In men, sTfR levels were inversely and significantly associated with waist circumference values and glucose levels, and positively with associated with LDL-C levels, before and after adjustments for confounding variables (Table 2). Also in men, LDL cholesterol had a significant and positive association with TAC (Table 3). Since TAC was positively associated either with LDL-C levels as sTfR levels, we conducted an additional multivariate regression analysis adding TAC as covariate for the significant association between LDL-C levels and sTfR levels mentioned above, and this remained significant (β = 11.66 confidence interval 95%: 4.84 to 18.49, *p* = 0.001).

In women, there was no significant relationship of variables of cardiometabolic risk with sTfR (Table 2) or TAC (Table 3).

Analyses were performed on transformed values of skewed variables: Logarithm of sTfR, ferritin, triglycerides, SBP and HOMA-IR values, and 1/CRP levels.

Significant relationships are shown in bold (*p* < 0.05).

Analyses were performed on transformed values of skewed variables: Logarithm of triglycerides, SBP and HOMA-IR values, and 1/CRP levels.

Significant relationships are shown in bold (*p* < 0.05).

## 4. Discussion

In this study, the relationship of TAC, a marker of antioxidant status, and cardiome- tabolic risk variables, with sTfR as an iron metabolism marker, was evaluated. TAC was a significant positive determinant of sTfR, independently of covariates. We also found that circulating sTfR concentrations were inversely associated with waist circumference. Moreover, these associations were not affected by ferritin and hepdicin levels among other adjustments in multivariate models.

Low sTfR levels reflect high iron stores. Therefore, the direction of the relationship with TAC suggests lower TAC in the presence of high iron status, probably as a consequence of iron-induced oxidative stress. Iron induces free radical production via Fenton and Haber-Ways reactions [6] while oxidative stress has also been found to modulate iron proteins. For instance, cell surface transferrin receptors are down-regulated in response to oxidants such as hydrogen peroxide, apparently as a compensatory mechanism in order to decrease additional oxidative damage in the cell that occurs following iron uptake [15]. In addition, in human proximal tubular epithelial cells, lower levels of protein TfR1 have been found when oxidative stress sensor NrF2 is downregulated by the inhibitor trigone- lline [16]. Meanwhile, at hepatic level, this effect of decreasing TfR1 levels was not observed in KO mice lacking of NrF2 gene [17].

We cannot discard the involvement of an alternative pattern: Higher sTfR in relation to higher TAC might imply iron deficiency and increased oxidative stress. In this way, higher TAC would represent as a compensatory mechanism to deal with high concentrations of free radicals. Oxidative stress not only might exhaust antioxidant defense but stimulate it. While it is less common, increased TAC levels have been reported in uncomplicated patients with type 2 diabetes in comparison with controls, and these patients also had higher levels of oxidized molecules (lipid peroxides) [18]. Both, low iron status and increased oxidative stress have been found associated with non-communicable chronic dieases (NCCD) and mortality. Levels of oxidative stress markers such as isoprostane (8-iso) and oxidized guanine/guanosine were found to be predictors of an epigenetic mortality risk score which in turn is highly associated with all-cause mortality [19]. Meanwhile, higher sTfR levels, a marker of iron deficiency or lower iron status, have been linked to prevalent cardiovascular disease in adult population from the NHANNES 2017–2018., and with all-cause mortality in healthy adult females also from USA [20,21]. Iron deficiency has been associated with higher levels of oxidized species such as malondialdehyde [22]. Particularly, in iron deficiency anaemia (IDA), oxidative stress can be highly increased because of premature death of red blood cells and subsequent release of pro-oxidant iron [23]. Similarly, RBC are more susceptible to oxidative damage in IDA [23].

In few studies conducted so far, the relationship between sTfR and cardiometabolic risk is inconsistent. In the present research sTfR levels were inversely associated with waist circumference and glucose suggesting increased iron stores with central obesity and associated metabolic alterations. The association with waist circumference is in line with the inverse correlation between levels of sTfR and visfatin, an adipokine mainly released by visceral fat [24]. In contrast, Montonen et al. [9] and Freixenet et al. [25] reported positive associations of sTfR with waist circumference. While Montonen et al. did not discuss this finding (their central outcome was prediction of type 2 diabetes), Freixenet et al. explained the higher sTfR levels in abdominal obesity of their hyperferritinemic subjects, in terms of increased infiltration of visceral fat by macrophages, which, unlike other cell types, increase their transferrin receptors as their cellular iron levels increase. Additionally, a population study in a Croatian population did not find associations between sTfR and components of the metabolic syndrome [10]. Unlike our sample, the Croatian study included subjects with chronic diseases and smoking habit, although these conditions were used as covariates. The authors claimed differences in the sTfR measurement techniques as a potential factor for the inconsistent link between sTfR levels and cardiometabolic risk across epidemiological studies [10].

Meanwhile, previous evidence of co-localization of transferrin receptors with insulin-responsive glucose transporters and IGF-II receptors in the microsomal membranes of adipocytes [26], are consistent with the inverse relationship of sTfR with fasting glucose observed in our subjects. However, the association between sTfR and LDL-C was strongly positive and persisted after controlling for parameters of chronic/subclinical inflammation, insulin resistance and other iron markers (ferritin and hepcidin), even after additional adjustment for TAC. This positive association suggests that, by still unknown mechanisms, cholesterol metabolism would be related to sTfR in the context of cardiometabolic risk. A recent study performed an exploratory factor analysis to examine the potential clustering of variables known to be associated with coronary artery disease using data from patients with angiographically-proved disease. Variables belonging to the inflammatory and obesity clusters predicted high ferritin values while the proatherogenic cluster factors (positive loadings of TC, LDL-C, apoB and TG) predicted high sTfR values [27]. In this study, the authors also compared the ability of the “proatherogenic factors” with that of a multivariable logistic model that included the “proatherogenic factor” and sTfR values in predicting significant stenosis in patients. The area under the ROC curve was 0.692 vs. 0.821, respectively. Further information is needed to corroborate this hypothesis.

Our findings of associations between sTfR and only very few variables of cardiometabolic risk are consistent with previous epidemiological studies that have reported very specific associations or none. For instance, a population study in a Croatian population, did not find independent associations between sTfR levels and components of metabolic syndrome, while several associations with serum ferritin were reported [10].

TAC was specifically and positively associated with LDL cholesterol levels, i.e., a higher antioxidant status according to higher levels of LDL-C. While LDL-C is implicated in the formation of atherogenic plaque via oxidation of components such as phospholipids in this lipoproteins, LDL-C is a complex particle with sub-fractions containing antioxidant properties, [28] such as paraoxonase-1 activity suggesting a possible defense mechanism from oxidation in LDL-C [29]. LDL-C is also known to carry several antioxidant vitamins such as Vitamin A and E [30]. The positive association of TAC with LDL-C could be due to the up-regulation of some antioxidant components in serum in response to LDL-C oxidation. Others studies found positive correlations of TAC with WC [30,31] and TG [31] in healthy individuals, whereas no correlations with hydroperoxide levels, an oxidative stress derived product, were described [32].

Current findings of significant relationships in men but not in women suggests a sexual dimorphism in the association between sTfR and cardiometabolic risk factors. Since men are known to exhibit higher iron status (lower sTfR) than women, relationships between cardiometabolic risk could be more evident in men above a threshold of iron stores. However, if it is iron status what is behind the associations between sTfR and WC and glucose, it is not clear why serum ferritin and/or hepcidin levels did not attenuate at all these associations. We previously reported in this same sample of subjects, positive associations between ferritin levels and insulin resistance (HOMA-IR index) and between hepcidin levels and systolic pressure and triglycerides [3,12]. This observation apparently would mean that iron markers can be associated with cardiometabolic risk via mechanisms other than iron metabolism. The effects of inflammation and/or infection on iron markers synthesis are well-known, but this effect would not explain the associations of sTfR found since adjustment for CRP levels did not attenuate them. Oxidative stress also up-regulates ferritin and hepcidin synthesis [33,34]. Particularly, ferritin appears to be linked to endothelial remodeling [35]. However, evidence of alternative factors influencing sTfR is scarce but there is high potential to find them in future in vitro research.

Our findings merit some warnings on the basis of limitations we must acknowledge. First, the study lacks of available measurement of hemoglobin levels to complement the characterization of iron status in the individuals. Since high sTfR levels reflect low bone marrow iron stores, this marker would also be indicative of erythropoiesis [36]. If so, in cases with anaemia or with trend to lower hemoglobin, a weak or null relationship of this marker with cardiometabolic risk variables would be expected, but it is unknown to what extent this pattern was present in our sample. Second, the cross-sectional design of our study does not enable inference about causal relationships. In addition, data regarding dietary iron intake would allow more robust adjustments. Likewise, the power of the study could have been limited by the fact of a heterogeneous female group in terms of premenopausal and postmenopausal women, and larger samples within each one of these subgroups will be required. The findings of this exploratory analysis may not be generalized before replication in large, multicenter studies. However, the strengths of this study include the use of a well characterized group of participants and use of a broad set of covariates for multivariate analyses.

Future research should focus on characterizing the antioxidant status in terms of not only TAC but also markers of increased oxidative stress. Moreover, the predicting capacity and trajectories of TAC with regard to the risk of non-communicable chronic diseases must be deeply explored in prospective studies since high levels of TAC, which mainly indicate a suitable antioxidant defense status, might also indicate a constant stimulus of TAC because of increased oxidative stress.

## 5. Conclusions

In summary, sTfR levels were associated with antioxidant status and with various cardio-metabolic risk markers independently of BMI, chronic inflammation, insulin resistance and other iron markers. Further research is required to further investigate the possible mechanisms of the relationship among iron metabolism, anti-oxidant status and several metabolic risk factors.

## Figures and Tables

**Figure 1 antioxidants-12-00019-f001:**
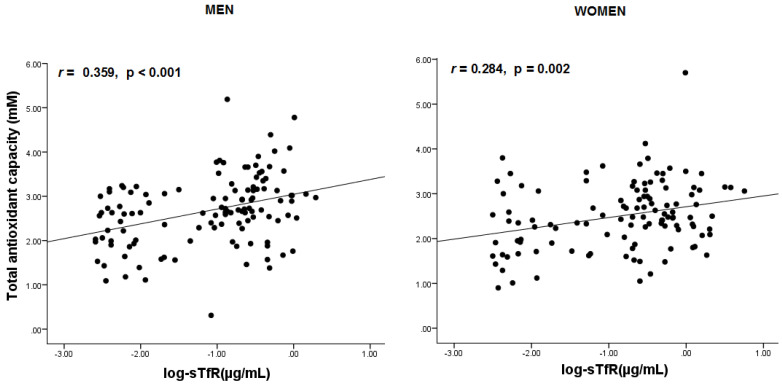
Correlations between levels of total antioxidant capacity and iron markers by gender. In subsequent multivariate linear regression analyses, the relationship between sTfR levels and TAC remained significant after adjusting age, menopause (for women), CRP levels, BMI, alcohol intake (no/yes), physical inactivity (no/yes), HOMA-IR, hepcidin and ferritin levels, either in men(β= 0.31 [0.14 to 0.48], *p* < 0.001) and women (β= 0.24 [0.07 to 0.40], *p* = 0.004). Analyses were performed on transformed values of skewed variables: Logarithm of sTfR, ferritin and and HOMA-IR values, and 1/CRP levels in the model.

**Table 1 antioxidants-12-00019-t001:** Description of the study population.

	Men	Women	*p* Value
n	122	117	
Age (years)	44.9 ± 7.7	46.1 ± 7.7	0.226
Menopause (n)		48	
BMI (kg/mts^2^)	26.2 ± 3.4	25.9 ± 3.7	0.501
sTfR (µg/mL)	0.42 (0.12–0.63)	0.54 (0.17–0.78)	0.014
Total antioxidant capacity (mM)	2.67 ± 0.78	2.50 ± 0.75	0.092
Waist circumference (cm)	85.1 ± 10.3	75.5 ± 8.0	<0.001
Systolic blood pressure (mmHg)	120 (118.8–129.1)	109.5 (101.5–120.8)	<0.001
Diastolic blood pressure (mmHg)	75.2 ± 8.8	72.1 ± 11.2	0.017
Triglycerides (mg/dL)	163 (117.7–228)	105.5 (80.2–154.2)	<0.001
Glucose (mg/dL)	91.4 ± 10.4	86.1 ± 8.3	<0.001
HDL-C (mg/dL)	44.0 ± 8.8	53.4 ± 11.5	<0.001
LDL-C (mg/dL)	113.1 ± 29.3	117.4 ± 28.9	0.251
hs-CRP (mg/L)	1.4 (1.1–1.9)	1.6 (1.1–3.0)	0.009
Ferritin (µg/L)	181 (128–269)	69.5 (23.5–120.7)	<0.001
Insulin (mU/mL)	9.46 (5.99–13.97)	7.47 (5.69–11.42)	0.043
Hepcidin (ng/mL)	1.23 (0.81–1.89)	0.65 (0.43–1.23)	<0.001
HOMA-IR	2.02 (1.26–2.99)	1.47 (1.05–2.43)	0.009

Data are mean ± standard deviation or median (interquartile range). BMI, body mass index. sTfR, soluble transferrin receptor. WC, waist circumference. HDL-C, HDL cholesterol. LDL-C, LDL cholesterol. hs-CRP, high sensitivity C reactive protein. HOMA-IR, homeostatic model assessment insulin resistance.

**Table 2 antioxidants-12-00019-t002:** βeta coefficients(95% confidence interval) for relationships between soluble transferrin receptor levels (sTfR) and variables of cardiometabolic risk.

	log-sTfR (µg/mL)	
	Non-Adjusted	Adjusted *
*Men*		
WC (cm)	−2.48 (−4.64 to −0.32)	−1.12 (−2.30 to −0.22)
Glucose (mg/dL)	−3.07 (−5.24 to −0.90)	−2.7 (−4.82 to −0.57)
HDL-C (mg/dL)	−1.02 (−2.89 to 0.84)	−1.70 (−3.48 to 0.07)
Log-Triglycerides (mg/dL)	−0.02 (−0.13 to 0.09)	0.02 (−0.07 to 0.13)
DBP (mmHg)	−0.04 (−1.92 to 1.83)	0.40 (−1.52 to 2.33)
log-SBP (mmHg)	−0.007 (−0.04 to 0.03)	−0.004 (−0.04 to 0.03)
LDL-C (mg/dL)	13.41 (7.54 to 19.28)	12.41 (6.08 to 18.57)

*Women*		
WC (cm)	−1.64 (−3.28 to 0.003)	−1.03 (−1.88 to −0.18)
Glucose (mg/dL)	−1.44 (−3.15 to 0.26)	−1.36 (−2.97 to 0.23)
HDL-C (mg/dL)	−0.35 (−2.76 to 2.04)	−0.58 (−2.95 to 1.78)
Log-Triglycerides (mg/dL)	−0.08 (−0.19 to 0.01)	−0.05 (−0.15 to 0.03)
DBP (mmHg)	0.71 (−1.62 to 3.05)	0.23 (−1.78 to 2.24)
log-SBP (mmHg)	−0.01 (−0.04 to 0.01)	−0.008 (−0.03 to 0.01)
LDL-C (mg/dL)	0.83 (−5.24 to 6.91)	2.94 (−3.26 to 9.15)

* Adjusted for: age, menopause (for women), CRP levels, BMI, alcohol intake (no/yes), physical inactivity (no/yes), HOMA-IR, hepcidin and ferritin levels.

**Table 3 antioxidants-12-00019-t003:** βeta coefficients (95% confidence interval) for relationships between serum total antioxidant capacity (TAC) with variables of cardiometabolic risk.

	TAC (mM)	
	Non-Adjusted	Adjusted *
*Men*		
WC (cm)	−1.31 (−3.71 to 1.08)	−0.88 (−1.83 to 0.06)
Glucose (mg/dL)	−1.09 (−3.53 to 1.34)	−0.67 (−2.94 to 1.59)
HDL-C (mg/dL)	0.97 (−1.07 to 3.02)	0.47 (−1.41 to 2.36)
Log-Triglycerides (mg/dL)	−0.07 (−0.19 to 0.05)	−0.06 (−0.16 to 0.04)
DBP (mmHg)	−0.23 (−2.26 to 1.78)	0.28 (−170 to 2.26)
log-SBP (mmHg)	0.01 (−0.02 to 0.06)	0.02 (−0.02 to 0.06)
LDL-C (mg/dL)	8.93 (2.13 to 15.73)	9.13 (2.30 to 15.97)

*Women*		
WC (cm)	−0.25 (−2.32 to 1.81)	−0.91 (−1.93 to 0.10)
Glucose (mg/dL)	0.07 (−2.02 to 2.18)	−1.49 (−3.34 to 0.35)
HDL-C (mg/dL)	−0.65 (−3.55 to 2.24)	0.07 (−2.80 to 2.95)
Log-Triglycerides (mg/dL)	0.02 (−0.10 to 0.14)	−0.03 (−0.15 to 0.08)
DBP (mmHg)	0.16 (−2.61 to 2.93)	−1.70 (−4.02 to 0.61)
log-SBP (mmHg)	−0.007 (−0.04 to 0.02)	−0.02 (−0.05 to 0.005)
LDL-C (mg/dL)	1.34 (−5.78 to 8.47)	0.10 (−7.37 to 7.57)

* Adjusted for: age, menopause (for women), CRP levels, BMI, alcohol intake (no/yes), physical inactivity (no/yes) and HOMA-IR.

## Data Availability

Not applicable.

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
