# Peer review of "Soluble Transferrin Receptor, Antioxidant Status and Cardiometabolic Risk in Apparently Healthy Individuals"

_antioxidants, 2022, doi:10.3390/antiox12010019_

Round 1
Reviewer 1 Report
The manuscript by Suarez-Ortegon et al." Soluble Transferrin Receptor, Antioxidant Status and Cardiometabolic Risk in Apparently Healthy Individuals" evaluated the relationship between cardiometabolic parameters in the general population and healthy individuals. The study addresses an important question. As previous studies pointed out, soluble transferrin receptor emerges as a good prediction factor for iron status, but its relationship with other health parameters needs to be evaluated. The authors described associations between levels of serum soluble transferrin, antioxidant status, and cardiometabolic risk variables.
Novel funding from this work could interest researchers in the iron and antioxidant field. Although the manuscript addresses a significant problem, it has some weaknesses and flaws to address. Here are major and minor comments.
Abstract
Line 21. Please include an explanation of the abbreviation TAC. It is explained later in the text but not in the abstract.
Introduction
Line 44. Please include a reference describing the association between ferritin and metabolic syndrome.
Line 49. There are several studies on sTfr as a marker of iron status. It would be more accurate to mention more previous studies and not only the publication from authors (ref.7).
Methods
Line 112. The authors explain the principle of total antioxidant capacity measurement but do not mention the commercial kit. Please describe the method or give a reference to the manufacture's protocol.
Discussion
Line 202. "Iron induces free radical production via Fenton and Haber-Ways reactions [7]"
Reference 7 is a citation of the author's work and does 'not describe free radical production via the Fenton reaction. Please give the correct reference.
“7. Fernández-Real, J.M., Moreno, J.M., López-Bermejo, A., et al. Circulating soluble transferrin receptor according to glucose tol-334 erance status and insulin sensitivity. Diabetes Care 2007;30(3):604-8. “
Line 207. "In addition, low expression of hepatic sTfR, have been found in mice lacking the oxidative 207 stress sensor NrF2 [16]." "16.Tanaka, Y., Ikeda, T., Yamamoto, K., et al. Dysregulated expression of fatty acid oxidation enzymes and iron-regulatory genes 361 in livers of Nrf2-null mice. J. Gastroenterol. Hepatol. 2012;27(11):1711-7. "
This reference mentions only liver expression of Tfr1 but not soluble Tfr1 ( sTfr). This study does not report a statistically significant difference in hepatic expression of Tfr1 in Nrf1 null and wild-typemice. Please correct.
Line 247. "As Us, a significant sTfR-WC association a population 247 study in a Croatian population did not find independent associations between sTfR levels and components of metabolic syndrome. However, the study population included population." This statement sounds unclear and might be a typo. Please clarify.
Line 274. "Proteins involved in iron metabolism might be differentially influenced by physiological states other than iron status or could participate in additional physiological functions." This statement is unclear and appears out of the contest. Please clarify.
Author Response
The manuscript by Suarez-Ortegon et al." Soluble Transferrin Receptor, Antioxidant Status and Cardiometabolic Risk in Apparently Healthy Individuals" evaluated the relationship between cardiometabolic parameters in the general population and healthy individuals. The study addresses an important question. As previous studies pointed out, soluble transferrin receptor emerges as a good prediction factor for iron status, but its relationship with other health parameters needs to be evaluated. The authors described associations between levels of serum soluble transferrin, antioxidant status, and cardiometabolic risk variables.
Novel funding from this work could interest researchers in the iron and antioxidant field. Although the manuscript addresses a significant problem, it has some weaknesses and flaws to address. Here are major and minor comments.
We thank the reviewer for the revision and the feed-back.
Abstract
Line 21. Please include an explanation of the abbreviation TAC. It is explained later in the text but not in the abstract.
R/ Done accordingly:
“Linear regressions adjusting for BMI, menopausal status, insulin resistance (HOMA-IR), physical inactivity, alcohol intake and subclinical/chronic inflammation were used to describe the association between sTfR, total antioxidant capacity (TAC), and measures of cardio-metabolic risk.”
Introduction
Line 44. Please include a reference describing the association between ferritin and metabolic syndrome.
R/We have inserted the following reference:
“Although some studies have identified associations between serum ferritin and the metabolic syndrome[4], less is known about the relationship between other markers of iron metabolism….”
Line 49. There are several studies on sTfr as a marker of iron status. It would be more accurate to mention more previous studies and not only the publication from authors (ref.7).
R/We have replaced the previous reference by a reference of classic study that describe iron metabolism and the way iron makers behave in relation to iron statuses of low and high demand. The new references listed is:
“7. Ganz T, Nemeth E. Iron imports. IV. Hepcidin and regulation of body iron metabolism. Am J Physiol Gastrointest Liver Physiol. 2006;290(2):G199-203. “
Methods
Line 112. The authors explain the principle of total antioxidant capacity measurement but do not mention the commercial kit. Please describe the method or give a reference to the manufacture's protocol.
R/ In this revised version we have provided the TAC assay manufacture`s catalog, as follows:
“Total antioxidant capacity (TAC) was measured by using a colorimetric assay (Cayman chemical, Cat No.709001, Ann Arbor MI) which assesses the total antioxidant capacity of serum in terms of the ability of ability of antioxidants in the sample to inhibit the oxida-tion of ABTS® (2,2'-azino-di-[3-ethylbenzthiazoline sulphonate]) to ABTS® · + by metmyoglobin. The capacity of the serum antioxidants is compared with Trolox, a to-copherol analogue, and quantified as molar Trolox equivalents.”
Discussion
Line 202. "Iron induces free radical production via Fenton and Haber-Ways reactions [7]"
Reference 7 is a citation of the author's work and does 'not describe free radical production via the Fenton reaction. Please give the correct reference.
“7. Fernández-Real, J.M., Moreno, J.M., López-Bermejo, A., et al. Circulating soluble transferrin receptor according to glucose tol-334 erance status and insulin sensitivity. Diabetes Care 2007;30(3):604-8. “
R/ We have replaced this wrong reference by the correct one, also previously mentioned in introduction section:
"Iron induces free radical production via Fenton and Haber-Ways reactions [6]"
6 .Fernández-Real, J.M., López-Bermejo, A., Ricart, W. Cross-talk between iron metabolism and diabetes. Diabetes 2002;51(8):2348-54.
Line 207. "In addition, low expression of hepatic sTfR, have been found in mice lacking the oxidative 207 stress sensor NrF2 [16]." "16.Tanaka, Y., Ikeda, T., Yamamoto, K., et al. Dysregulated expression of fatty acid oxidation enzymes and iron-regulatory genes 361 in livers of Nrf2-null mice. J. Gastroenterol. Hepatol. 2012;27(11):1711-7. "
This reference mentions only liver expression of Tfr1 but not soluble Tfr1 ( sTfr). This study does not report a statistically significant difference in hepatic expression of Tfr1 in Nrf1 null and wild-typemice. Please correct.
R/ Apologies for this mistake. In the same section of discussion, we have added a study which showed significant reduction of TRF1 by down regulation of NrF2 in renal tubular cells, and we have contrasted this finding by describing the non-significant findings in the mice study of Tanaka et al. :
“Low sTfR levels reflect high iron stores. Therefore, the direction of the relationship with TAC suggests lower TAC in the presence of high iron status, probably as a conse-quence of iron-induced oxidative stress. Iron induces free radical production via Fenton and Haber-Ways reactions [6] while oxidative stress has also been found to modulate iron proteins. For instance, cell surface transferrin receptors are down-regulated in response to oxidants such as hydrogen peroxide, apparently as a compensatory mechanism in order to decrease additional oxidative damage in the cell that occurs following iron uptake [15]. In addition, in human proximal tubular epithelial cells, lower level of protein TfR1 have been found when oxidative stress sensor NrF2 is downregulated by the inhibitor trigonel-lin [16]. On the other hand, at hepatic level, this effect of decreasing TfR1 levels was not observed in KO mice lacking of NrF2 gene [17].”
- Van Raaij SE, Masereeuw R, Swinkels DW, van Swelm RP. Inhibition of Nrf2 alters cell stress induced by chronic iron exposure in human proximal tubular epithelial cells. Toxicology Letters. 2018;295:179-86.
- Tanaka, Y., Ikeda, T., Yamamoto, K., et al. Dysregulated expression of fatty acid oxidation enzymes and iron-regulatory genes in livers of Nrf2-null mice. J. Gastroenterol. Hepatol. 2012;27(11):1711-7.
Line 247. "As Us, a significant sTfR-WC association a population 247 study in a Croatian population did not find independent associations between sTfR levels and components of metabolic syndrome. However, the study population included population." This statement sounds unclear and might be a typo. Please clarify.
R/ Yes, indeed was a typo mistake. We have corrected as follows:
“Our findings of associations between sTfR and only very few variables of cardiometabolic risk are consistent with previous epidemiological studies that have reported very specific associations or none. For instance, a population study in a Croatian population, did not find independent associations between sTfR levels and components of metabolic syndrome, while several associations with serum ferritin were reported.”
Line 274. "Proteins involved in iron metabolism might be differentially influenced by physiological states other than iron status or could participate in additional physiological functions." This statement is unclear and appears out of the contest. Please clarify.
R/ We totally agree with the reviewer. The sentence is vague and confusing. Thus, we have decided to remove it from the manuscript, since the idea of the paragraph in which it was previously inserted, does not change after removing it:
“Current findings of significant relationships in men but not in women suggests a sexual dimorphism in the association between sTfR and cardiometabolic risk factors. Since men are known to exhibit higher iron status (lower sTfR) than women, relationships between cardiometabolic risk could be more evident in men above a threshold of iron stores. However, if it is iron status what is behind the associations between sTfR and WC and glucose, it is not clear why serum ferritin and/or hepcidin levels did not attenuate at all these associations. We previously reported in this same sample of subjects, positive associations between ferritin levels and insulin resistance (HOMA-IR index) and between hepcidin levels and systolic pressure and triglycerides [3, 12]. This observation apparently would mean that iron markers can be associated with cardiometabolic risk via mechanisms other than iron metabolism. The effects of inflammation and/or infection on iron markers synthesis are well-known, but this effect would not explain the associations of sTfR found since adjustment for CRP levels did not attenuate them. Oxidative stress also up-regulates ferritin and hepcidin synthesis [26, 27]. Particularly, ferritin appears to be linked to endothelial remodeling [28]. However, evidence of alternative factors influencing sTfR is scarce but there is high potential to find them in future in vitro research.”
Reviewer 2 Report
The manuscript entitled: Soluble transferrin receptor, antioxidant status and cardiometabolic risk in apparently healthy individuals by Dr Suares-Ortegon et al. with corresponding author Professor Fernández-Real reports on joint work between departments of the Facultad de Ciencias de La Salud, Pontificia Universidad Jave- 6 riana Seccional Cali, Cali, Colombia, departments of the Universidad del Valle, Cali, Colombia, and the department of Diabetes, Endocrinology and Nutrition, Institut d'Investigació Biomèdica de Girona (IdIBGi), 10 CIBEROBN (CB06/03/010) and the Instituto de Salud Carlos III (ISCIII), Girona, Spain.
Dr Suares-Ortegon performed academic research in Scotland for some time and is an experienced investigator. Professor Fernandez-Real is an authority in his field. Together they are at the basis of the study design and together with senior author Professor M. Mosquera they assured the funding of the research. They have significant publication records. They also have a long-term record of publishing papers in collaboration.
This scientific level and experience of the authors can be recognized in the manuscript that is written in good English, needing no more than slight editing. (an example: there is an unfinished sentence on lines 249-250). Moreover, the manuscript, notably the introduction and discussion sections, have been written in a clear and educational way. It is rewarding to read such a document where every line demonstrates important knowledge and insight acquired during many years of research in complex fields including those of metabolic disease and iron metabolism. There is a significant extent of self-citation in the references section but in view of the educational character of the paper this becomes functional and is not disturbing.
On the other hand, relatively few recent articles by other authors have been quoted. This leaves the reader, aware of recent literature with more questions than answers. Moreover, it leads to a lack of equilibrium between recent and older references. Including appropriate recent references and discussing the relevance of their findings in the context of the recent literature would allow the authors to illustrate the place of their results even better in the context of today’s state of knowledge in this field.
I give examples:
a) The authors demonstrated an association between sTfR levels and the antioxidant status in apparently healthy individuals. This is an interesting finding that could exactly fit in the scope of the journal. However, the relevance of this total antioxidant capacity for (predicting) health issues has not been very clearly discussed. Oxidative stress has been associated with mortality in several studies (e.g. 1) and is believed to play a role in aging. sTfR levels also have been associated with cardiovascular disease (2, 3) and even with all-cause mortality in apparently healthy individuals. (4) It would be highly appropriate if the authors would give their in-depth views on the apparent contradictory findings: High sTfR with high total antioxidant capacity and high mortality? It seems important to compare the characteristics of the group of individuals involved in the study of the authors and that in (2 and especially 4). Can we compare the range of sTfR levels between the groups and learn from that? Are we forced to envisage a phenomenon comparable to the adiponectin paradox where many preclinical studies predict a favorable impact on health and yet high levels are associated with increased mortality? Would a high total antioxidant capacity, so far considered a salutary parameter, in some conditions be counterproductive and if so, when and why? Can we exclude this possibility? Discussing these issues would also form an appropriate warning to readers.
Would there be an alternative explanation?
b) The sTfR was found to be associated with the total antioxidant capacity. A high sTfR level implies, according to the authors, low bone marrow iron stores and thus less iron overload related oxidative stress. However, as the authors state in one of their other papers, (their reference 10) quoting the paper of Beguin 2003, in this cohort of apparently healthy people and in the age group selected, sTfR is expected to be indicative of erythropoiesis. This may explain the inverse association between levels of sTfR and hemoglobin found in controls by others. (e.g. 5) It is therefore regrettable that the authors did not provide and discuss the hemoglobin and serum iron levels that were probably available. A recent paper reports a U-shaped association of hemoglobin concentration levels with metabolic syndrome and metabolic components. (6) It would be very illustrative to see in which part of the hemoglobin curve the individuals in the cohort studied by the authors are located. Could there also be a non-linear association between hemoglobin and the total antioxidant capacity, so that TAC would be highest at the hemoglobin levels where the odds ratios of insulin resistance are at their lowest values?
This comparison using hemoglobin levels may show (or not) why there is no association with most components of the metabolic syndrome but with insulin resistance (their work in reference 10). This obviously will require the authors to also discuss the recent view that oxidative stress cannot only find its origin in iron overload but also in iron deficiency. (7).
The authors demonstrated an excellent capacity to analyze the literature in-depth.
For these reasons I advise the authors:
a) to (also) discuss the association of iron deficiency with oxidative stress.
b) To discuss the above-mentioned issues.
c) To provide an adequate warning for the reported association between high levels of soluble transferrin receptor and mortality even though the authors found it to be associated with a high total antioxidant capacity.
d) to extend their list of references with, I guess a dozen recent relevant papers. This will probably be required for these tasks. The references provided form examples.
I believe that this can lead to a manuscript that would be more appropriate in view of the scope of Antioxidants. I congratulate the authors with their careers and wish them success with a possible revision.
References
1) Oxidative stress and epigenetic mortality risk score: associations with all-cause mortality among elderly people.
Gao X, Gào X, Zhang Y, Holleczek B, Schöttker B, Brenner H. Eur J Epidemiol. 2019 May;34(5):451-462. doi: 10.1007/s10654-019-00493-7. Epub 2019 Feb 15. PMID: 30771035
2) Increased Serum Soluble Transferrin Receptor Levels Were Associated With High Prevalence of Cardiovascular Diseases: Insights From the National Health and Nutrition Examination Survey 2017-2018.
Zhu S, Liu C, Zhao C, Chen G, Meng S, Hong M, Xiang M, Xie Y.Front Cell Dev Biol. 2022 Apr 12;10:874846. doi: 10.3389/fcell.2022.874846. eCollection 2022.PMID: 35493097
3) High soluble transferrin receptor in patients with heart failure: a measure of iron deficiency and a strong predictor of mortality.
Sierpinski R, Josiak K, Suchocki T, Wojtas-Polc K, Mazur G, Butrym A, Rozentryt P, van der Meer P, Comin-Colet J, von Haehling S, Kosmala W, Przewlocka-Kosmala M, Banasiak W, Nowak J, Voors AA, Anker SD, Cleland JGF, Ponikowski P, Jankowska EA. Eur J Heart Fail. 2021 Jun;23(6):919-932. doi: 10.1002/ejhf.2036. Epub 2020 Dec 4.PMID: 33111457
4) Soluble transferrin receptor can predict all-cause mortality regardless of anaemia and iron storage status. Kang M, Kwon S, Lee W, Kim Y, Bae E, Lee J, Park JY, Kim YC, Kim EY, Kim DK, Lim CS, Kim YS, Lee JP. Sci Rep. 2022 Jul 13;12(1):11911. doi: 10.1038/s41598-022-15674-w.PMID: 35831434
5) Serum transferrin receptor concentration indicates increased erythropoiesis in Kenyan children with asymptomatic malaria.
Verhoef H, West CE, Ndeto P, Burema J, Beguin Y, Kok FJ.Am J Clin Nutr. 2001 Dec;74(6):767-75. doi: 10.1093/ajcn/74.6.767.PMID: 11722958
6) Positive or U-Shaped Association of Elevated Hemoglobin Concentration Levels with Metabolic Syndrome and Metabolic Components: Findings from Taiwan Biobank and UK Biobank.
Timoteo VJ, Chiang KM, Pan WH. Nutrients. 2022 Sep 27;14(19):4007. doi: 10.3390/nu14194007.PMID: 36235661
7) Iron deficiency, immunology, and colorectal cancer.
Phipps O, Brookes MJ, Al-Hassi HO. Nutr Rev. 2021 Jan 1;79(1):88-97. doi: 10.1093/nutrit/nuaa040.PMID: 32679587 Review.
Author Response
The manuscript entitled: Soluble transferrin receptor, antioxidant status and cardiometabolic risk in apparently healthy individuals by Dr Suares-Ortegon et al. with corresponding author Professor Fernández-Real reports on joint work between departments of the Facultad de Ciencias de La Salud, Pontificia Universidad Jave- 6 riana Seccional Cali, Cali, Colombia, departments of the Universidad del Valle, Cali, Colombia, and the department of Diabetes, Endocrinology and Nutrition, Institut d'Investigació Biomèdica de Girona (IdIBGi), 10 CIBEROBN (CB06/03/010) and the Instituto de Salud Carlos III (ISCIII), Girona, Spain.
Dr Suares-Ortegon performed academic research in Scotland for some time and is an experienced investigator. Professor Fernandez-Real is an authority in his field. Together they are at the basis of the study design and together with senior author Professor M. Mosquera they assured the funding of the research. They have significant publication records. They also have a long-term record of publishing papers in collaboration.
This scientific level and experience of the authors can be recognized in the manuscript that is written in good English, needing no more than slight editing. (an example: there is an unfinished sentence on lines 249-250). Moreover, the manuscript, notably the introduction and discussion sections, have been written in a clear and educational way. It is rewarding to read such a document where every line demonstrates important knowledge and insight acquired during many years of research in complex fields including those of metabolic disease and iron metabolism. There is a significant extent of self-citation in the references section but in view of the educational character of the paper this becomes functional and is not disturbing.
On the other hand, relatively few recent articles by other authors have been quoted. This leaves the reader, aware of recent literature with more questions than answers. Moreover, it leads to a lack of equilibrium between recent and older references. Including appropriate recent references and discussing the relevance of their findings in the context of the recent literature would allow the authors to illustrate the place of their results even better in the context of today’s state of knowledge in this field.
I give examples:
- a)The authors demonstrated an association between sTfR levels and the antioxidant status in apparently healthy individuals. This is an interesting finding that could exactly fit in the scope of the journal. However, the relevance of this total antioxidant capacity for (predicting) health issues has not been very clearly discussed. Oxidative stress has been associated with mortality in several studies (e.g. 1) and is believed to play a role in aging. sTfR levels also have been associated with cardiovascular disease (2, 3) and even with all-cause mortality in apparently healthy individuals. (4) It would be highly appropriate if the authors would give their in-depth views on the apparent contradictory findings: High sTfR with high total antioxidant capacity and high mortality? It seems important to compare the characteristics of the group of individuals involved in the study of the authors and that in (2 and especially 4). Can we compare the range of sTfR levels between the groups and learn from that? Are we forced to envisage a phenomenon comparable to the adiponectin paradox where many preclinical studies predict a favorable impact on health and yet high levels are associated with increased mortality? Would a high total antioxidant capacity, so far considered a salutary parameter, in some conditions be counterproductive and if so, when and why? Can we exclude this possibility? Discussing these issues would also form an appropriate warning to readers.
Would there be an alternative explanation?
R/ We thanks so much the reviewer for this highly pertinent point of view. Indeed, we should talk about the alternative association pattern that means a positive association sTfR-TAC. In this revised version we have added sentences discussing an alternative meaning of iron deficiency and oxidative stress in terms of stimulated TAC as follows:
“We cannot discard the involvement of an alternative pattern: higher sTfR in relation to higher TAC might imply iron deficiency and increased oxidative stress. In this way, higher TAC would represent as a compensatory mechanism to deal with high concentrations of free radicals. Oxidative stress not only might exhaust antioxidant defense but also stimulate it. Although it is less common, increased TAC levels have been reported in uncomplicated patients with type 2 diabetes in comparison with controls, and these patients also had higher levels of oxidized molecules (lipid peroxides)[18]. Both, low iron status and increased oxidative stress have been found associated with non-communicable chronic diseases (NCCD) and mortality. Levels of oxidative stress markers such as isoprostane (8-iso) and oxidized guanine/guanosine were found to be predictors of an epigenetic mortality risk score which in turn is highly associated with all-cause mortality [19]. Meanwhile, higher sTfR levels, a marker of iron deficiency or lower iron status, have been linked to prevalent cardiovascular disease in adult population from the NHANNES 2017-2018., and with all-cause mortality in healthy adult females also from U.S [20, 21]. Iron deficiency has been associated with higher levels of oxidized species such as malondialdehyde [22]. Particularly, in iron deficiency anaemia (IDA), oxidative stress can be highly increased because of premature death of red blood cells and subsequent release of pro-oxidant iron [23]. Similarly, RBC are more susceptible to oxidative damage in IDA [23].”
In respect to the paradox of TAC about whether high levels would also be a predictor of disease, we have added this concern as a perspective for future research at the end of discussion section as follows:
“Future research should focus on characterize antioxidant status in terms of not only TAC but also markers of increased oxidative stress. Moreover, the predicting capability and trajectories of TAC with regard the risk of non-communicable chronic diseases must deeply explored in prospective studies since high levels of TAC, which mainly indicate a suitable antioxidant defense status, might also indicate a constant stimulus of TAC because of increased oxidative stress.”
We could not compare ranges of sTfR levels in the studies provided by the reviewer (#2 and #4) because one of these articles only provided values log-transformed of sTfR levels.
- b)The sTfR was found to be associated with the total antioxidant capacity. A high sTfR level implies, according to the authors, low bone marrow iron stores and thus less iron overload related oxidative stress. However, as the authors state in one of their other papers, (their reference 10) quoting the paper of Beguin 2003, in this cohort of apparently healthy people and in the age group selected, sTfR is expected to be indicative of erythropoiesis. This may explain the inverse association between levels of sTfR and hemoglobin found in controls by others. (e.g. 5) It is therefore regrettable that the authors did not provide and discuss the hemoglobin and serum iron levels that were probably available. A recent paper reports a U-shaped association of hemoglobin concentration levels with metabolic syndrome and metabolic components. (6) It would be very illustrative to see in which part of the hemoglobin curve the individuals in the cohort studied by the authors are located. Could there also be a non-linear association between hemoglobin and the total antioxidant capacity, so that TAC would be highest at the hemoglobin levels where the odds ratios of insulin resistance are at their lowest values?
This comparison using hemoglobin levels may show (or not) why there is no association with most components of the metabolic syndrome but with insulin resistance (their work in reference 10). This obviously will require the authors to also discuss the recent view that oxidative stress cannot only find its origin in iron overload but also in iron deficiency. (7).
R/ The issue on hemoglobin levels as complementary information has been addressed as a limitation of our study (the study lacked of Hemoglobin and serum iron measurements), as follows:
“Our findings merit some warnings on the basis of limitations we must acknowledge. First, the study lacks of available measurement of hemoglobin levels to complement the characterization of iron status in the individuals. Since high sTfR levels reflect low bone marrow iron stores, this marker would also be indicative of erythropoiesis [36]. If so, in cases with anaemia or with trend to lower hemoglobin, a weak or null relationship of this marker with cardiometabolic risk variables would be expected, but it is unknown to what extent this pattern was present in our sample.”
The authors demonstrated an excellent capacity to analyze the literature in-depth.
For these reasons I advise the authors:
- a)to (also) discuss the association of iron deficiency with oxidative stress.
R/ This discussion has been done as part of the alternative explanation for the sTfR-TAC relationship, as follows:
“We cannot discard the involvement of an alternative pattern: higher sTfR in relation to higher TAC might imply iron deficiency and increased oxidative stress. In this way, higher TAC would represent as a compensatory mechanism to deal with high concentrations of free radicals. Oxidative stress not only might exhaust antioxidant defense but stimulate it. Although it is less common, increased TAC levels have been reported in uncomplicated patients with type 2 diabetes in comparison with controls, and these patients also had higher levels of oxidized molecules (lipid peroxides)[18]. Both, low iron status and increased oxidative stress have been found associated with non-communicable chronic diseases (NCCD) and mortality. Levels of oxidative stress markers such as isoprostane (8-iso) and oxidized guanine/guanosine were found to be predictors of an epigenetic mortality risk score which in turn is highly associated with all-cause mortality [19]. Meanwhile, higher sTfR levels, a marker of iron deficiency or lower iron status, have been linked to prevalent cardiovascular disease in adult population from the NHANNES 2017-2018., and with all-cause mortality in healthy adult females also from U.S [20, 21]. Iron deficiency has been associated with higher levels of oxidized species such as malondialdehyde [22]. Particularly, in iron deficiency anaemia (IDA), oxidative stress can be highly increased because of premature death of red blood cells and subsequent release of pro-oxidant iron [23]. Similarly, RBC are more susceptible to oxidative damage in IDA [23].”
- b)To discuss the above-mentioned issues.
R/ The issues about alternative explanation sTfR-TAC relationship, availability of haemoglobin variable, and TAC paradox, have been addressed in the following two sections of discussion section, as follows:
alternative explanation sTfR-TAC relationship
“We cannot discard the involvement of an alternative pattern: higher sTfR in relation to higher TAC might imply iron deficiency and increased oxidative stress. In this way, higher TAC would represent as a compensatory mechanism to deal with high concentrations of free radicals. Oxidative stress not only might exhaust antioxidant defense but stimulate it. Although it is less common, increased TAC levels have been reported in uncomplicated patients with type 2 diabetes in comparison with controls, and these patients also had higher levels of oxidized molecules (lipid peroxides)[18]. Both, low iron status and increased oxidative stress have been found associated with non-communicable chronic diseases (NCCD) and mortality. Levels of oxidative stress markers such as isoprostane (8-iso) and oxidized guanine/guanosine were found to be predictors of an epigenetic mortality risk score which in turn is highly associated with all-cause mortality [19]. Meanwhile, higher sTfR levels, a marker of iron deficiency or lower iron status, have been linked to prevalent cardiovascular disease in adult population from the NHANNES 2017-2018., and with all-cause mortality in healthy adult females also from U.S [20, 21]. Iron deficiency has been associated with higher levels of oxidized species such as malondialdehyde [22]. Particularly, in iron deficiency anaemia (IDA), oxidative stress can be highly increased because of premature death of red blood cells and subsequent release of pro-oxidant iron [23]. Similarly, RBC are more susceptible to oxidative damage in IDA [23].”
availability of haemoglobin variable
“Our findings merit some warnings on the basis of limitations we must acknowledge. First, the study lacks of available measurement of hemoglobin levels to complement the characterization of iron status in the individuals. Since high sTfR levels reflect low bone marrow iron stores, this marker would also be indicative of erythropoiesis [36]. If so, in cases with anaemia or with trend to lower hemoglobin, a weak or null relationship of this marker with cardiometabolic risk variables would be expected, but it is unknown to what extent this pattern was present in our sample.”
TAC paradox
“Future research should focus on characterize antioxidant status in terms of not only TAC but also markers of increased oxidative stress. Moreover, the predicting capacity and trajectories of TAC with regard the risk of non-communicable chronic diseases must deeply explored in prospective studies since high levels of TAC, which mainly indicate a suitable antioxidant defense status, might also indicate a constant stimulus of TAC be-cause of increased oxidative stress.”
- c)To provide an adequate warning for the reported association between high levels of soluble transferrin receptor and mortality even though the authors found it to be associated with a high total antioxidant capacity.
R/ This (c) comment of the reviewer is similar to the comment described in (a). In this revised version we have added sentences discussing an alternative meaning of iron deficiency and oxidative stress in terms of stimulated TAC as follows:
“We cannot discard the involvement of an alternative pattern: higher sTfR in relation to higher TAC might imply iron deficiency and increased oxidative stress. In this way, higher TAC would represent as a compensatory mechanism to deal with high concentrations of free radicals. Oxidative stress not only might exhaust antioxidant defense but stimulate it. Although it is less common, increased TAC levels have been reported in uncomplicated patients with type 2 diabetes in comparison with controls, and these pa-tients also had higher levels of oxidized molecules (lipid peroxides)[18]. Both, low iron status and increased oxidative stress have been found associated with non-communicable chronic diseases (NCCD) and mortality. Levels of oxidative stress markers such as isoprostane (8-iso) and oxidized guanine/guanosine were found to be predictors of an epigenetic mortality risk score which in turn is highly associated with all-cause mortality [19]. Meanwhile, higher sTfR levels, a marker of iron deficiency or lower iron status, have been linked to prevalent cardiovascular disease in adult population from the NHANNES 2017-2018., and with all-cause mortality in healthy adult females also from U.S [20, 21]. Iron deficiency has been associated with higher levels of oxidized species such as malondialdehyde [22]. Particularly, in iron deficiency anaemia (IDA), oxidative stress can be highly increased because of premature death of red blood cells and subsequent release of pro-oxidant iron [23]. Similarly, RBC are more susceptible to oxidative damage in IDA [23].”
- d)to extend their list of references with, I guess a dozen recent relevant papers. This will probably be required for these tasks. The references provided form examples.
R/ In this revised version, we have used most of the references kindly provided by the reviewer. See updated list of references. We used references 1, 2, 4 and 5. We were unable to use reference #7 of the reviewer since that paper did not provide any mention of mechanisms linking iron deficiency and oxidative stress. We used other reference for that link.
I believe that this can lead to a manuscript that would be more appropriate in view of the scope of Antioxidants. I congratulate the authors with their careers and wish them success with a possible revision.
Many thanks for all your comments and suggestions
References
1) Oxidative stress and epigenetic mortality risk score: associations with all-cause mortality among elderly people.
Gao X, Gào X, Zhang Y, Holleczek B, Schöttker B, Brenner H. Eur J Epidemiol. 2019 May;34(5):451-462. doi: 10.1007/s10654-019-00493-7. Epub 2019 Feb 15. PMID: 30771035
2) Increased Serum Soluble Transferrin Receptor Levels Were Associated With High Prevalence of Cardiovascular Diseases: Insights From the National Health and Nutrition Examination Survey 2017-2018.
Zhu S, Liu C, Zhao C, Chen G, Meng S, Hong M, Xiang M, Xie Y.Front Cell Dev Biol. 2022 Apr 12;10:874846. doi: 10.3389/fcell.2022.874846. eCollection 2022.PMID: 35493097
3) High soluble transferrin receptor in patients with heart failure: a measure of iron deficiency and a strong predictor of mortality.
Sierpinski R, Josiak K, Suchocki T, Wojtas-Polc K, Mazur G, Butrym A, Rozentryt P, van der Meer P, Comin-Colet J, von Haehling S, Kosmala W, Przewlocka-Kosmala M, Banasiak W, Nowak J, Voors AA, Anker SD, Cleland JGF, Ponikowski P, Jankowska EA. Eur J Heart Fail. 2021 Jun;23(6):919-932. doi: 10.1002/ejhf.2036. Epub 2020 Dec 4.PMID: 33111457
4) Soluble transferrin receptor can predict all-cause mortality regardless of anaemia and iron storage status. Kang M, Kwon S, Lee W, Kim Y, Bae E, Lee J, Park JY, Kim YC, Kim EY, Kim DK, Lim CS, Kim YS, Lee JP. Sci Rep. 2022 Jul 13;12(1):11911. doi: 10.1038/s41598-022-15674-w.PMID: 35831434
5) Serum transferrin receptor concentration indicates increased erythropoiesis in Kenyan children with asymptomatic malaria.
Verhoef H, West CE, Ndeto P, Burema J, Beguin Y, Kok FJ.Am J Clin Nutr. 2001 Dec;74(6):767-75. doi: 10.1093/ajcn/74.6.767.PMID: 11722958
6) Positive or U-Shaped Association of Elevated Hemoglobin Concentration Levels with Metabolic Syndrome and Metabolic Components: Findings from Taiwan Biobank and UK Biobank.
Timoteo VJ, Chiang KM, Pan WH. Nutrients. 2022 Sep 27;14(19):4007. doi: 10.3390/nu14194007.PMID: 36235661
7) Iron deficiency, immunology, and colorectal cancer.
Phipps O, Brookes MJ, Al-Hassi HO. Nutr Rev. 2021 Jan 1;79(1):88-97. doi: 10.1093/nutrit/nuaa040.PMID: 32679587 Review.
Round 2
Reviewer 1 Report
The authors of the manuscript Suarez-Ortegon et al." Soluble Transferrin Receptor, Antioxidant Status and Cardiometabolic Risk in Apparently Healthy Individuals" have addressed reviewers' comments and questions and revised the manuscript. I recommend publishing this work.
Reviewer 2 Report
The authors have responded to the different questions asked and to the suggestions made. They have clarified the issues raised. The manuscript is almost ready for publication. Some, often minor, English editing would still benefit the paper, particularly in the newly added text but to some extent also in the entire text. This would not change the scientific merit but it would benefit the readability and thereby benefit the reputation of both the authors and the journal.
A small issue: The text: "Body weight and hight .........(BMI)......"appears both in Clinical measurements and in Biochemical measurements.
PS
In a future study it could be very informative to measure oxidized LDL in the same study population.